# The Use of Autologous Cell Therapy in Diabetic Patients with Chronic Limb-Threatening Ischemia

**DOI:** 10.3390/ijms251810184

**Published:** 2024-09-23

**Authors:** Dominika Sojakova, Jitka Husakova, Vladimira Fejfarova, Andrea Nemcova, Radka Jarosikova, Simon Kopp, Veronika Lovasova, Edward B. Jude, Michal Dubsky

**Affiliations:** 1Diabetes Centre, Institute for Clinical and Experimental Medicine, 14021 Prague, Czech Republic; sojd@ikem.cz (D.S.); hadj@ikem.cz (J.H.); vlfe@ikem.cz (V.F.); nema@ikem.cz (A.N.); jaar@ikem.cz (R.J.); koxs@ikem.cz (S.K.); 2First Faculty of Medicine, Charles University, 14021 Prague, Czech Republic; 3Transplantation Surgery Department, Institute for Clinical and Experimental Medicine, 14021 Prague, Czech Republic; lovv@ikem.cz; 4Second Faculty of Medicine, Charles University, 15006 Prague, Czech Republic; 5Diabetes Center, Tameside and Glossop Integrated Care NHS Foundation Trust, Ashton-under-Lyne OL6 9RW, UK; edward.jude@tgh.nhs.uk; 6Department of Endocrinology and Gastroenterology, University of Manchester, Manchester M13 9PL, UK

**Keywords:** peripheral artery disease, chronic limb-threatening ischemia, stem cell therapy

## Abstract

Autologous cell therapy (ACT) is primarily used in diabetic patients with chronic limb-threatening ischemia (CLTI) who are not candidates for standard revascularization. According to current research, this therapy has been shown in some studies to be effective in improving ischemia parameters, decreasing the major amputation rate, and in foot ulcer healing. This review critically evaluates the efficacy of ACT in patients with no-option CLTI, discusses the use of mononuclear and mesenchymal stem cells, and compares the route of delivery of ACT. In addition to ACT, we also describe the use of new revascularization strategies, e.g., nanodiscs, microbeads, and epigenetics, that could enhance the therapeutic effect. The main aim is to summarize new findings on subcellular and molecular levels with the clinical aspects of ACT.

## 1. Introduction

Peripheral artery disease (PAD) and diabetic neuropathy are the main causes of diabetic foot disease, which leads to foot ulcerations and loss of limbs. The prevalence of PAD is 10–26% in the population without diabetes and 20–50% in people with diabetes [1]. Chronic limb-threatening ischemia (CLTI) is a clinical syndrome defined by the presence of PAD combined with rest pain, gangrene, or an ulcer on the lower limb present for least 2 weeks [2]. Typical symptoms of PAD include rest and claudication pain in the legs, an impaired quality of life, and ulcer formation, which lead to amputation or death [3]. However, intermittent claudication and rest pain may be absent in people with diabetes due to diabetic neuropathy. Arterial involvement in patients suffering from PAD and diabetes is usually bilateral, multisegmental, and infra-popliteal [4].

Standard strategies for the treatment of CLTI include percutaneous transluminal angioplasty (PTA) and a vascular bypass. When comparing these two treatment methods, the BEST-CLI study demonstrated a lower incidence of all-cause mortality and major adverse limb events (amputations above the ankle) in patients with an adequate great saphenous vein after a bypass compared to PTA [5]. Results were similar in patients who lacked an adequate saphenous vein conduit. On the other hand, the BASIL-2 study recommended PTA for people who required an infra-popliteal revascularization, as the first treatment approach had a longer amputation-free survival compared to the vein bypass [6]. Recently, there has been increased emphasis on angiosome-based revascularization, also known as direct revascularization. This is a method that improves the perfusion of lost tissue directly through its feeder artery, while the more traditional “best vessel approach”, known as indirect revascularization, involves the selection of the most appropriate target vessel for revascularization regardless of its anatomical location. Recent reviews suggest that direct revascularisation seems to be a reasonable approach in diabetic patients, who usually have poor collateralization and would probably benefit from revascularization directly into the feeding artery at the site of tissue loss [7].

## 2. When Standard Revascularization Is Not Possible

Approximately 25% of patients with CLTI are not eligible for revascularization (no-option CLTI) [8]. These patients also have high mortality and the worst prognosis, with a risk of death of approximately 20–25% within one year and around 60% within five years of being diagnosed with CLTI [9]. The prognosis is even worse in chronic kidney disease (CKD). The prevalence of CLTI in individuals with CKD is 24%. Patients with end-stage CKD and PAD had shorter overall survival times and higher rates of major amputations compared to those without kidney impairment or with CKD stages 1–3 [10].

A promising revascularization method is a treatment using extracorporeal shockwave therapy (ESWT), which helps increase blood flow to the limbs by stimulating the formation of collateral circulation and slowing the process of arteriosclerosis in the lower limbs. A meta-analysis published in 2023 that included four randomized controlled trials showed that ESWT is an effective treatment for patients with PAD, improving claudication pain, increasing claudication-free walking distance, and reducing stenosis in the affected limb; however, no significant improvement in the ankle–brachial index (ABI) was noted [11].

Another option is treatment with prostanoids, but the majority of the studies did not demonstrate an effect either on wound healing or major amputations [12,13]. However, in a recent study, alprostadil was used after IR; this adjuvant therapy significantly improved ulcer healing and pain relief compared to the control group and raised the proportion of patients who survived without major amputation during follow-up [14]. This suggests that the use of prostanoids in patients with CLTI has potential as an adjuvant treatment together with standard revascularization.

Finally, the transcatheter arterialization of deep veins is a possible treatment too. During the procedure on the lower limbs, an arteriovenous fistula is created proximal to the affected tibial arteries using a covered stent. Oxygenated blood is then diverted from the tibial arteries to the tibial veins to bypass the severely affected arterial vessel. The venous system is used to deliver oxygenated arterial blood to the leg via the pedal veins. The single-aim PROMISE II study assessed the effect of the procedure on amputation-free survival and limb salvage. After 6 months, 66.1% of patients achieved amputation-free survival and 76% of patients avoided major amputation. Therefore, it can be concluded that arterialization is safe and could be performed successfully in patients with CLTI [15].

## 3. New Promising Options for Revascularization

Recently, a number of studies have assessed the options for CLTI at the genetic or nanoparticle level. In this review, we mention only a few of them. Nanoceria-decorated graphene oxide (CeGO) represents a pathway for increasing blood flow and reduces the pro-inflammatory status in the CLTI region. After an injection into the ischemic hindlimb in mice, the CeGO-cell spheroids inhibit the accumulation of reactive oxygen species (ROS) that are overproduced in the ischemic leg area, but also preserve a high cell survival rate and increase angiogenic molecular expression, thus enhancing blood reperfusion and tissue recovery [16]. Other experiments with fibrin microbeads, which consist of human umbilical vein endothelial cells and bone marrow-derived MSCs showed an increased reperfusion of ischemic limbs and improved limb salvage via the formation of microvascular vessels [17]. The transmembrane form of stem cell factors delivered in lipid nanodiscs injected into the ischemic limbs of rabbits with hyperlipidemia and diabetes demonstrated higher vascularity in comparison to alginate-treated controls 8 weeks after injection [18]. Terbium hydroxide nanorods (THNRs) improved survival and promoted proliferation, migration, the restoration of nitric oxide production, and the regulation of vascular permeability in hypoxia-exposed endothelial cells, as well as a reduction in muscle damage and inflammation by stimulation of PI3K/AKT/eNOS and probably also Wnt/GSK-3β/β-catenin signaling pathways [19].

The research on the genetic level has already described epigenetic possibilities. These epigenetic techniques include DNA methylation, histone modifications, and actions of long non-coding RNAs (lncRNAs), which are defined as RNA transcripts in length that do not code proteins but play roles as regulatory molecules. In PAD, there has been experimental studies with the modulation of metastasis-associated lung adenocarcinoma transcript 1 (MALAT1), maternally expressed gene 3 (MEG3), antisense non-coding RNA in the INK4 locus (ANRIL), small nucleolar host gene 12 (SNHG12), and lncRNA [20]. All these new methods are currently experimental, but it is a challenge for the future to transfer them to clinical practice and thus extend the possibilities for no-option patients.

## 4. Stem Cell Possibilities in Ischemia Treatment

Another current treatment for no-option patients is stem cell therapy. There are several possibilities for choosing the type of cells, their processing, and routes of administration.

### 4.1. Type of Stem Cells

Commonly used cells for cell therapy are bone marrow-derived mononuclear cells (BM-MNCs) and peripheral blood mononuclear cells (PB-MNCs). In PB-MNCs, prior stimulation with either the granulocyte colony-stimulating factor (G-CSF) or the granulocyte-macrophage colony-stimulating factor (GM-CSF) is required. Mesenchymal stem cells (MSCs) are also widely used and they can be isolated from different sources, which include bone marrow, peripheral blood, and adipose tissue, as well as neonatal tissues (particular parts of the placenta and umbilical cord) [21]. They are easily accessible for separation and have a broad differentiation potential into other tissue types. They present low immunogenicity and therefore, it is possible to provide both autologous and allogeneic therapy, which can be a challenge when it comes to improving the treatment effect due to the possibility of selecting donors with fewer comorbidities. On the other hand, MSCs in tissue are low in both autologous and allogeneic therapy, so in vitro expansion is necessary [22].

A promising group of stem cells are endometrial-derived stem cells (EnSCs), which are divided into endometrial epithelial-like cells and endometrial stromal-like cells. Endometrial epithelial stem cells are very difficult to isolate and culture long-term in vitro due to their limited proliferation and functional capacity. Therefore, most current studies on endometrial stem cells focus on stromal endometrial stem cell types [23]. They have similar paracrine activity and lower levels of immunogenicity as MSCs. Moreover, they can differentiate into myoblasts or myocytes under controlled in vitro and in vivo conditions. Finally, the differentiation of EnSCs into neural cells has been confirmed under in vitro conditions. Therefore, using EnSCs could potentially not only stimulate angiogenesis in the affected limb but also improve the condition of muscle tissue and peripheral nerves [24].

Since most diabetic patients with CLTI also suffer from neuropathy, using neural stem cells (NSCs) could possibly achieve the regeneration of peripheral nerve damage in people with diabetic peripheral neuropathy [25]. However, there are no studies regarding the use of NSCs in the treatment of CLTI patients. The only published study on the use of NSCs in patients after a stroke had demonstrated some improvement in motor mobility in patients after ACT, but many such studies are needed [26]. All types of stem cells used in ACT or considered for this therapy are shown in Figure 1.

### 4.2. Stem Cell Subpopulations

The different types of stem cell subpopulations have been discussed in several studies, but its precise description remains unknown. Asahara et al., in 1997, were the first to use the term “putative endothelial cells”, which may be responsible for increased collateral vessel growth in ischemic tissues [27]. Since then, various cell subpopulations have been referred to as endothelial progenitor cells (EPCs). Using flow cytometry, EPCs are identified as a percentage of mononuclear cells expressing the cluster of differentiation 34 (CD34) and vascular endothelial growth factor receptor 2 (VEGFR2). These cells may also represent circulating mature endothelial cells from the vascular system, which is why CD133 has been added as another progenitor marker [28]. Finally, in 2017, Medina published an article where he divided EPCs into two cell lines, namely myeloid angiogenic cells (MACs) and endothelial colony-forming cells (ECFCs) [29].

#### 4.2.1. Myeloid Angiogenic Cells (MACs)

MACs are defined as cultured cells derived from peripheral mononuclear cells, identified by surface markers CD45, CD14, and CD31, as well as the absence of CD146 and CD31 markers. They cannot become endothelial cells, but they support angiogenesis through a paracrine mechanism [29]. MACs have been clinically used only for the treatment of myocardial infarction [30] and pulmonary arterial hypertension [31]. It has been shown that peripheral blood MACs are reduced in patients with diabetes, their mobilization is impaired, and vascular repair capacity is reduced too. MACs in people with diabetes also show higher inflammatory potential due to the upregulation of interleukin β (IL β), which has been identified as a key mediator for the exacerbation of ischemia [32]. Another mediator of change was thrombospondin-1 [33]. In a study testing anti-inflammatory treatment with canakinumab (an antibody targeting IL1b) for atherosclerotic disease has shown that canakinumab protects patients from recurrent cardiovascular events [34].

#### 4.2.2. Endothelial Colony-Forming Cells (ECFCs)

ECFCs represent endothelial cells with a strong intrinsic angiogenic capacity that are able to repair vascular endothelial damage and promote the formation of new blood vessels. They express surface markers CD31, CD105, and CD146, and lack markers CD146 and CD14 [29]. The frequency of markers is variable and may vary with in vitro cell expansion. They can be isolated from peripheral or cord umbilical blood, but it has been reported that ECFCs can be found in adipose tissue, placenta, and lungs as well. The unique role of ECFCs in the process of angiogenesis in cell therapy remains unclear, but there are several possibilities, i.e., the direct incorporation of ECFCs at sites of vascular injury, the paracrine effect mediated by the secretion of pro-angiogenic cells, and supporting the reparative capacity of other cells such as MSCs [35]. According to a preclinical study, the benefit of increasing ECFCs to improve perfusion by rapid transfer to ischemic mouse limbs has been demonstrated within 6 h after intravenous injection with ECFCs [36].

#### 4.2.3. Limitations of ECFCs and Options for Their Enhancement

There are several limitations to the use of ECFCs for ACT, most notably their low number in diabetes, which is even further reduced in older patients with diabetes, and the long cultivation time, which would prolong the overall duration of ACT [37]. Their frequency is very low in both cord blood and peripheral blood, with approximately 50 ECFCs to 1 × 10^8^ cord blood mononuclear cells and 1.7 PB-ECFCs per 1 × 10^8^ PB-MNCs. ECFCs isolated from the cord blood of patients with gestational diabetes showed increased senescence and decreased proliferation, migration, and tube formation compared to those in ECFCs from the cord blood of healthy pregnant females. In addition, it has been reported that it was more difficult to isolate ECFCs from the peripheral blood of type 2 diabetes patients than from healthy donors. Triggering factors that lead to ECFC dysfunction in patients with diabetes are hyperglycemia (via the PI3K/Akt/eNOS pathway, the decreased SIRT1 level, and the AGEs–RAGE axis), oxidative stress (increased ROS-in diabetic patients), and inflammation (increased levels of plasma inflammatory cytokines) [38]. Therefore, from a clinical point of view, the inflammation associated with chronic ulcers should be suppressed by antibiotics and we should aim for the best possible glycemic control before stem cell administration, thus increasing the effect of stem cell therapy.

The ability to improve ECFC function is further demonstrated in several studies. This was achieved by combining ECFCs and MSCs due to the fact that the ECFCs can operate via paracrine mediators, modulating MSC engraftment via (platelet-derived growth factor) PDGF-BB/PDGFRb signaling [39]. Another study confirmed potential benefits for vessel recovery with reduced inflammatory and necrotic processes when MSCs were used with ECFCs isolated from adipose tissue or with ECFCs from cord blood simultaneously [40,41]. MSCs increase the vasculogenic capacity of ECFCs by the formation of pericyte-covered perfused blood vessels, since ECFCs injected alone do not form these vessels [42]. There were also attempts to improve the effect of ECFCs by genetically modifying them to overexpress integrin β1, which would facilitate the return of ECFCs to ischemic tissue [43]. Fucoidan, a marine sulfated polysaccharide extracted from various species of algae or seaweed, restores the expression of functional ECFC surface markers (CD34, c-Kit, VEGFR2, and CXCR4) and stimulates the in vitro tube formation capacity of ECFCs [44]. The pre-treatment of ECFCs with ARA290 (an agonist of erythropoietin) enhanced the proliferation, migration, and resistance to H2O2-induced apoptosis of ECFCs [45]. The glycomimetic C3 also increases tube formation in ECFCs from diabetic patients with neuro-ischemic ulcers [46]. Moreover, it was demonstrated that vitamin D has a protective effect on ECFCs in vitro and also enhances their migration and cell-to-cell adhesion by increasing endothelial interconnections through vascular endothelial cadherin (VE-cadherin) junctions and by impacting cell dynamics through cofilin and VE-cadherin phosphorylation [47]. Thrombin receptor PAR1 silencing in ECFCs significantly enhances post-ischemic revascularization in a mouse model of hindlimb ischemia by inducing cell proliferation, evidenced by the silencing of the receptor PAR1 and followed by the overexpression of Ki67 [48]. Recent studies suggest that miR-134-5p may serve as a novel target for improving dysfunctional diabetic ECFCs, because the inhibition of miR-139-5p inhibits the c-jun-VEGF/PDGF-B pathway and thus can reverse the impaired migration and tubule formation ability of diabetic ECFCs [49].

#### 4.2.4. Other Cell Types and Markers

Patel et al. published a study which focused on the role of monocytes in ACT, specifically circulating proangiogenic and arteriogenic monocytes expressing the CD16 marker, which showed enhanced large vessel remodeling in the injected muscle area of five patients in the study. Moreover, their secretion of angiogenic factors, the vascular endothelial growth factor (VEGF-A), and the heparin-binding epidermal growth factor was demonstrated [50].

New markers have been identified that demonstrated endothelial vasoreparative cells. For example, CD157 has recently been identified as a tissue marker of vascular endothelial stem cells in many mouse organs. Mouse vascular endothelial stem cells (CD45CD31+CD157+) showed increased endothelial tubule formation and demonstrated clonal endothelial colony formation properties compared to the CD157 cell population. Nevertheless, the existence of these markers has not yet been proven in humans. Other promising types of stem cells are endothelial cells expressing the endothelial protein C receptor (EPCR), which contributes to endothelial cell expansion during mammary vasculature expansion and has the ability to differentiate into pericytes [51]. It is almost certain that other types of stem cells will be introduced in the future, as well as the possibility of enhancing their effect, for example, through their combined administration or prestimulation with other agents. Although all of the above methods are potential treatments for CLTI patients, the question remains of which of them will be included into clinical practice.

## 5. Cell Therapy Process in Clinical Practice

### 5.1. Inclusion and Exclusion Criteria

Our center has been providing this treatment for more than 15 years. Criteria for the application of stem cells in our institute include a diagnosis of diabetes, age of 18–90 years, the presence of diabetic foot ulcers or gangrene, CLTI (transcutaneous oxygen tension—TcPO_2_—below 30 mmHg), and the inability to undergo standard revascularization (a percutaneous transluminal angioplasty, bypass, or endarterectomy). The unsuitability of standard revascularization procedures is determined by a multidisciplinary team composed of a diabetologist, interventional radiologist, and vascular surgeon. Contraindications for cell therapy include severe limb edema, serious hematological abnormalities, untreated severe diabetic retinopathy, cancer of any organ, and the inability to undergo general anesthesia and a history of deep vein thrombosis, myocardial infarction, or stroke within the last 6 months. Different studies had variable criteria for administering ACT. For example, the PROVASA study, except for infra-popliteal occlusion, also included patients with ischemia in the femoropopliteal region [52]. Other studies excluded patients with major tissue loss and patients on renal replacement therapy, but on the other hand, they enrolled subjects without diabetes [53]. Also, the use of immunosuppressive therapy is very often an exclusion criterion too [54].

### 5.2. Use of Mononuclear Stem Cells

The stem cell therapy process consists of the collection of stem cells (SCs), processing of the SCs in a specialized sterilized laboratory, and their injection into the patient. We use BM-MNCs, which are obtained by a trephine biopsy of the iliac bone crest. Bone marrow products are isolated using either the Harvest Smart PReP2 device (Harvest Technologies Corporation, Plymouth, MA, USA) or a Gelofusine solution, followed by further centrifugation with a benchtop centrifuge (Hettich ROTANTA 460R, Andreas Hettich GmbH & Co. KG, Tuttlingen, Germany). The second source of stem cells, which we used in our institute, was PB-MNCs. All these patients had a central venous catheter inserted. PB-MNCs were separated by leukapheresis on Haemonetics MCS+ using 16–20 cycles for the achievement of a minimal concentration of CD34+ cells in peripheral blood 2 × 10^4^/mL after stimulation by 5–8 mg/kg/day of G-CSF for previously 3–6 days [55].

After the isolation of either BM-MNCs or PB-MNCs, 40–50 mL of the final product was delivered intramuscularly into the patient’s calf muscle and around the ulcer area (approximately 30–40 injections) in the operating room. During the procedure, spinal, epidural, or general anesthesia was used depending on the preferences of the anesthesiologist and the patient.

### 5.3. Mesenchymal Stem Cells in ACT

MSCs can also be used, but their concentrations in tissue are low; therefore, in vitro expansion is required for their subsequent therapeutic use. The isolation methods vary depending on their source. In a study using BM-MSCs, bone marrow was purified by density gradient centrifugation, and the resulting mononuclear cell (MNC) fraction was seeded in a culture medium in flasks and subsequently cryopreserved until injection [56]. For therapy with WJ-MSC processing therapy, small pieces of the umbilical cord were placed in a collagenase solution, followed by incubation in a solution of trypsin and ethylenediaminetetraacetic acid (EDTA). After dissolving the mononuclear cells in minimum essential medium eagle (MEM), they were seeded with periodic medium changes. After one week, the cells were trypsinized and washed. After reaching 70–80% confluence, they were detached with trypsin and EDTA. Finally, the cells were expanded in vitro to produce the necessary amount for treatment [54].

The procedure depends on the cell type and source of cells. Despite this, every stem cell institute has its own adapted protocol in accordance with their technical and organizational possibilities. Similarly, the price of therapy depends on the chosen method, but in accordance with our experience, it is comparable to the price of PTA.

The process with both MNCs and MSCs is schematically illustrated in Figure 2.

## 6. Routes of Stem Cell Administration

In addition to intramuscular cell delivery, there are other techniques of cell injections. The advantage of the intramuscular administration of precursor cells is the possibility to inject cells directly near the location of the ulcer, which stimulate paracrine activity and the release of angiogenic cytokines such as VEGF, bFGF (basic fibroblast growth factor), placental growth factor (PIGF), and monocyte chemoattractant protein-1 (MCP-1) after treating with MSCs; and VEGF, bFGF, hepatocyte growth factor (HGF), and angiopoietin-1 using MNCs [22,57]. Furthermore, intramuscular injection is less invasive and harmful, which is preferable in CLTI patients. The main purpose of the intra-arterial method is to direct stem cells to peripheral areas with sufficient oxygen and nutrient supply to promote the activity of new cells [57]. Intra-arterial administration increases the risk of arterial injury, including dissection of the vessel wall and dislocation of atherosclerotic plaques [58]. A study published by Klepanec et al. did not show a significant difference in limb preservation and ulcer healing between intramuscular and intra-arterial stem cell delivery [59]. The disadvantage of the local administration of cells, both intramuscularly and intra-arterially, is their early apoptosis. In intravenous systemic administration, there is a significant first-pass effect through the lung, where some of them remain and therefore, a higher dose of cells injected is needed to reach the ischemia location [60]. The systemic administration of stem cells is less common because of the difficulty of the administration and adverse events. In our opinion, it is very questionable if the stem cells administered via the systemic route are able to transfer into the ischemic location with significant effects.

## 7. Treatment Results

There are several outcomes by which we can assess the effect of ACT. One of the most common surrogate endpoints is limb salvage or amputation-free survival. Despite this, several studies describe their impact on wound healing, as well as the improvement of ischemia parameters, ischemic pain, and quality of life.

### 7.1. Selected Studies Supporting the Effect of Stem Cell Therapy

According to a meta-analysis by Sun et al., ACT significantly reduces the rate of major amputations and the time to ulcer healing, improves ischemia parameters, and extends the distance without claudication pain compared to standard therapy [61]. A recent meta-analysis involving 12 randomized studies demonstrated the benefits of ACT in reducing the incidence of major amputations, alleviating rest pain, and improving ischemia parameters [62]. A study by Meloni et al. found that only 16% of patients underwent major amputation within a year following ACT. It is important to note that this study used PB-MNCs and these were administered three times [63]. Dubsky et al. compared the effect of therapy between patients treated with PTA, ACT, or a placebo, in which more healed ulcers were observed in the ACT group than in the PTA and placebo groups. However, the difference in amputation-free survival was greater in patients treated by ACT and PTA versus the placebo, without significant differences between the active treatment groups [64].

### 7.2. Studies Comparing Different Stem Cell Types

Regarding the comparison of the success of therapy using BM-MNCs versus PB-MNCs, our previous study did not demonstrate any difference between these two types of stem cells [55]. However, an important moment in the use of PB-MNCs is the choice of prestimulation. It has been reported that G-CSF stimulation is more effective, but individual studies are influenced by differences in ethnicity and the presence of comorbidities [57]. Another recent study compared the efficacy of BM-MNCs versus allogeneic MSCs derived from Wharton’s jelly in patients with diabetic CLTI and found that patients treated with both types of stem cells avoided major amputation, but 60% of patients in the placebo group underwent amputation during the 12-month follow-up. Furthermore, the therapeutic benefit in terms of TcPO_2_ was significantly shorter in patients following MSC application [54]. A study published by Lu et al. in 2011 showed more healed ulcers in a shorter time after the application of MSCs from bone marrow compared to the application of BM-MNCs. There was also an improvement in perfusion as measured by TcPO_2_, ABI, walking distance without claudication, and a reduction in stenoses demonstrated by magnetic resonance angiography. On the other hand, there was no observed difference in the number of amputations and pain reduction [65]. Personally, in terms of the process, invasiveness, and results, we would recommend the use of MSCs to BM-MNCs because of the invasiveness and the lack of the option of allogeneic therapy using source tissue from a donor as in MSCs.

### 7.3. In-Depth Analysis of the Most Important Studies

We have chosen five randomized studies that, in our opinion, have resonated the most of out of all the publications dealing with this issue so far. The main characteristics of all critical studies are shown in Table 1.

#### 7.3.1. PROVASA

PROVASA was a multicenter, phase II, double-blind, randomized-start trial published in 2011 that included 40 patients with CLTI who received either intra-arterial BM-MNCs or a placebo followed by active treatment with BM-MNCs (open-label) after 3 months [52]. ABI was used as the main ischemia parameter in this study and was not significantly different between ACT and placebo groups. We believe that it was a wrong choice of primary outcome because ABI is strongly affected by medial sclerosis in diabetic patients. On the other hand, the secondary parameters, TcPO_2_, ulcer healing, and a reduction in rest pain, were significantly improved in the ACT group compared to the placebo. This is another reason for including TcPO_2_ as a main measurement in the analysis of patients with CLTI, especially after ACT. PROVASA had a very interesting crossover design—at the end of 3 months, all patients who had received a placebo switched to an active BM-MNC treatment, and patients who had initially received BM-MNCs received a repetitive treatment. Autologous serum was used in this study as a placebo. The main issues in this study were ABI as the primary endpoint and the intra-arterial administration of BM-MNCs, which in our opinion is inferior to intramuscular injections.

#### 7.3.2. JUVENTAS

JUVENTAS was a randomized, double-blind, placebo-controlled trial published in 2015 that evaluated the efficacy of BM-MNCs compared to a placebo [66]. This study included 160 patients; the cell suspension of BM-MNCs was applied intra-arterially in three doses within 1 week, and the placebo group received a suspension of erythrocytes (similar color). JUVENTAS showed no significant difference in survival or major amputation rates or ischemia parameters (ABI and TcPO_2_) between ACT and placebo groups. These outcomes were in contrast with the results of the other studies, and this discrepancy could be explained especially by the definition of CLTI—the mean baseline TcPO_2_ in JUVENTAS was quite high (35 ± 22 mm Hg); therefore, the potential of ACT to improve CLTI was limited. ABI, on the other hand, could be influenced by a high number of diabetic patients in this study (37.5%) and presence of medial sclerosis. Another important issue could be the patency in the aorto–femoro–popliteal area in JUVENTAS; 58.8% of patients had significant stenoses and occlusions in this area, which could negatively impact the effect of ACT. In this study, only 100 mL of bone marrow was harvested and only 1/3 was injected immediately, and the rest was frozen and used later. There could also be an influence of residual platelets in the placebo, which contained erythrocytes for color reasons.

#### 7.3.3. Study by Sharma et al.

A study published by Sharma et al. in 2020 was a randomized, double-blind, placebo-controlled trial to evaluate the safety and efficacy of intra-arterial BM-MNC treatment in 56 patients with CLTI and 25 patients with severe claudication [67]. The study showed a significant increase in ABI and TcPO_2_, and significantly fewer amputated patients compared to the placebo. The placebo was described as the patient’s own serum with peripheral blood erythrocytes to match the color of the cell suspension. Limitations of this study were the enrollment not only of CLTI patients but also people with claudication and a relatively small sample size.

#### 7.3.4. RESTORE-CLI

RESTORE-CLI was a randomized, double-blind, placebo-controlled, multicenter study that was performed at 18 centers in the United States in 86 patients with no-option CLTI [68]. This well-designed study showed a longer time to treatment failure (which was a composite endpoint defined as major amputation, death, doubling of the size of the wound, and newly formed gangrene) in the ACT group compared with the placebo. The study also showed significantly more healed ulcers in the ACT group within 12 months. The placebo group in this study was well defined even with the sham at the iliac crest but without penetration of the periosteum. The cell suspension in RESTORE-CLI contained tissue repair cells defined as a mixture of nucleated cells cultured from the patient’s bone marrow with high viability. This suspension was injected into the muscles of ischemic limbs. Even though this study is quite old, published in 2011, it is in accordance with our own results [69] and we believe that including a composite endpoint as primary endpoint improved the strength of the study.

#### 7.3.5. PACE

PACE is a very recent study, published in 2024; it is a randomized, controlled, multicenter, multinational, phase III study that compared treatment with mesenchymal allogeneic cells (PLX-PAD—PLacental eXpanded) versus a placebo [53]. The study enrolled 213 patients; 143 were randomized to receive PLX-PAD and 70 to the placebo. PACE showed no significant difference in amputation rates, amputation-free survival, or healing rates between PLX-PAD and the placebo. However, the beneficial effects of PLX-PAD treatment were noted in patients without diabetes or those with well-controlled diabetes, i.e., HbA1c levels below 6.5%. The placebo in this study contained dimethyl sulphoxide, human serum, albumin, and PlasmaLyte. The main problem of this large study was that the definition of CLTI was too high—an ankle pressure below 70 mm Hg and toe pressure below 50 mm Hg is not a definition of CLTI. We believe that the severity of CLTI patients in the PACE study is similar to the JUVENTAS study—both these studies included patients with milder ischemia and not real no-option CLTI and therefore, the outcomes could be affected. From this, it may be concluded that ACT could be effective in people with the most severe stages of limb ischemia, whereas its efficacy in those with mild or moderate ischemia is substantially lower.

#### 7.3.6. Summary of Key Studies

To summarize all the key studies, it can be can stated that stem cell therapy has a potential as a treatment for patients with CLTI, but the efficacy of ACT in clinical trials depends on the study design, inclusion and exclusion criteria, and endpoints. In the PACE or JUVENTAS study, patients were enrolled with high values of ankle and toe pressure or TcPO_2_. The Sharma trial included people with claudication without ulcers. These criteria do not meet the definition of CLTI. The primary endpoint of the PROVASA study was ABI, that is, the inexact parameter in patients with diabetes because of medial sclerosis. Furthermore, the differences in the volume of injected suspension also impacted the effect of therapy. In accordance with the design of the JUVENTAS trial, these patients were treated with a smaller amount of stem cells. As we mentioned above, intramuscular routes of injection potentially contribute to better results for their local paracrine effects. Hence, this may have caused poorer results in studies with the intra-arterial administration of stem cells. Finally, regarding the control group, it depends on the choice of placebo, because using autologous serum could potentially influence the results of cell therapy; even small amounts of autologous cells can potentially improve the placebo effect.

## 8. Is ACT a Safe Method?

A meta-analysis focusing on the safety of ACT, in addition to its efficacy, concluded that cellular therapy is a safe method for no-option CLTI patients using BM-MNCs, stimulated PB-MNCs, MSCs, or even cultured mononuclear cells [70]. Murphy et al., in a study with BM-MNCs, observed only in two patients ST depressions without the elevation of cardiac-specific markers, accompanied by a decrease in hemoglobin, which normalized after a transfusion, and a microembolism, which was successfully managed surgically [71]. As adverse events during ACT at our clinic, we observed bleeding after the trepanation of bone marrow and the temporary worsening of limb edema after injection [69]. It is important to mention that PB-MNC therapy involves less invasive collection and processing, which is associated with fewer adverse effects. Nonetheless, administering G-CSF for PB-MNC use also has side effects, such as bone pain, headaches, fatigue, and “flu-like syndrome” [72]. Severe cases have rarely been described, such as splenic rupture [73], neutrophil-induced pulmonary toxicity [74], and transient hypercoagulable state [75].

To focus on MSC administration, results of several studies indicated that MSC therapy was safe [76,77]. Moon et al. noted cellulitis on untargeted sites, paresthesia, uncontrolled diabetes, and cardiac arrest; however, none of these serious adverse events were considered to be related to the treatment [78]. In accordance with global vascular guidelines from 2019, it is a recommended therapy on level 1B, so it is possible to consider this therapy safe [79].

## 9. Conclusions

In conclusion, ACT is a promising treatment for patients with CLTI who cannot undergo standard revascularization. There are several research opportunities in this field for the future. Firstly, the selection of the most effective stem cell type for patients must be researched further. Regarding the fragility and comorbidities of CLTI patients, mesenchymal cell therapy, specifically allogeneic, seems to be the most effective method with the least invasiveness. In our opinion, the best route of the subsequent administration is intramuscular, because of the possibility of injection near the ulcer and connected paracrine effects. Another challenge is increasing the effect of ACT by enhancing cell subpopulations, specifically ECFCs, but we think that this progress will probably remain at the experimental level as more data and experience are collected.

## Figures and Tables

**Figure 1 ijms-25-10184-f001:**
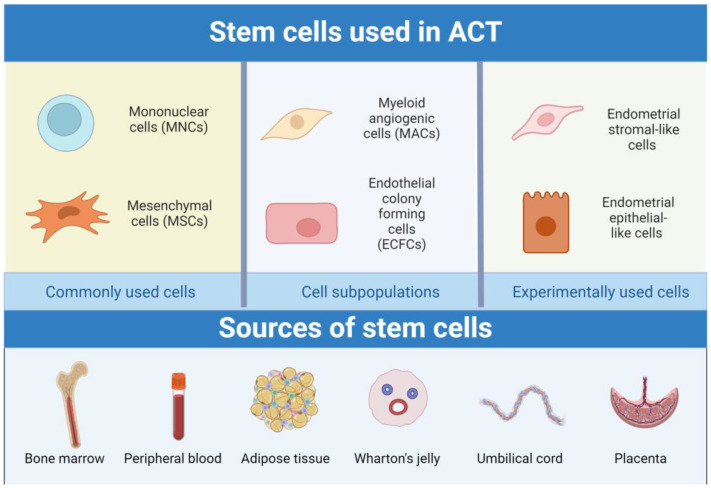
Types of stem cells used in ACT are divided into a group of commonly used cells, cell subpopulations, and experimentally used cells.

**Figure 2 ijms-25-10184-f002:**
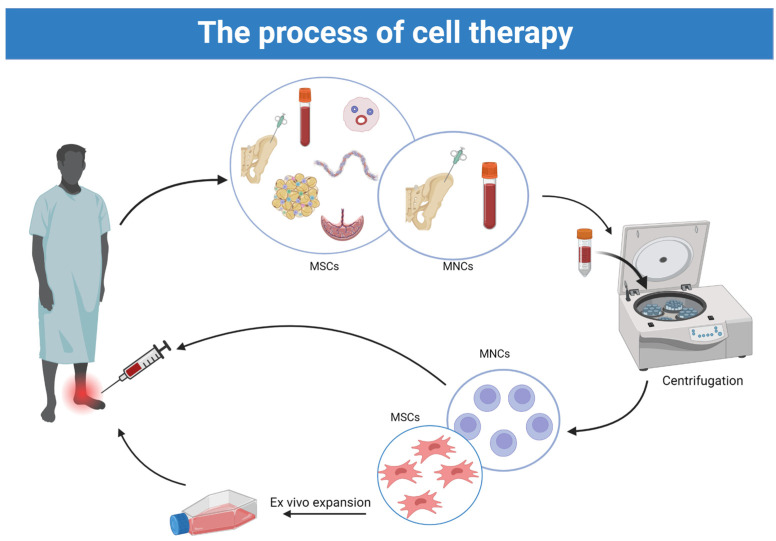
Cell therapy process using mononuclear stem cells (MNCs) and mesenchymal stem cells (MSCs).

**Table 1 ijms-25-10184-t001:** Overview of important studies with their design and main findings.

Study, Author, Year	Number of Patients	Cell Types	Control Group	Routes of Injection	Main Findings
PROVASA, Walter DH et al., 2011 [52]	40	BM-MNCs	Autologous serum	IA	No significance in ABI, but TcPO_2_, ulcer healing, and reduction in rest pain were significantly improved.
JUVENTAS, Teraa M et al., 2015 [66]	160	BM-MNCs	Suspension of erythrocytes (similar color)	IA	No significance in survival or major amputation rates or ischemia parameters (ABI, TcPO_2_).
Sharma S et al., 2021 [67]	81	BM-MNCs	Patient’s own serum with peripheral blood erythrocytes	IA	Significant increase in ABI and TcPO_2_ and significantly fewer amputated patients.
RESTORE-CLI, Powel RJ et al., 2011 [68]	46	Tissue repair cells	Electrolyte solution without cells	IM	Longer time to major amputation, death, doubling of the size of the wound, and newly formed gangrene. Significantly more healed ulcers within 12 months.
PACE, Norgren L et al., 2024 [53]	213	Mesenchymal allogeneic cells (PLX-PAD—PLacental eXpanded)	Suspension of dimethyl sulphoxide, human serum, albumin, and PlasmaLyte	IM	No significant difference in amputation rates, amputation-free survival, or healing rates. The beneficial effects were noted in patients without diabetes or those with well-controlled diabetes, i.e., HbA1c levels below 6.5%.

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
