# Peer review of "The Use of Autologous Cell Therapy in Diabetic Patients with Chronic Limb-Threatening Ischemia"

_ijms, 2024, doi:10.3390/ijms251810184_

Round 1

Reviewer 1 Report

Comments and Suggestions for Authors

The manuscript needs some rearrangement and more clear and better description of findings, and their sumarry. Here are some remarks.
Major remarks:
1.    Fig. 1 would be more informative presenting known ways of CLTI treatment instead of mentioning possible cell types.
2.    The paragraphs 4 and 5 needs connection with further separation into subsections (4.1….; 4.2…) dedicated for each cell types used in the CLTI. The information also should be summarized in Table 1 with the citations identifying cell type, ways of application, final result, progress or other main findings important for the CLTI therapy.
3.     The section 5 is mainly only about the one cells type, ECFC. What about MAC cells? Only at the end some other cells were mentioned. The section title does not correspond to the content and is not clearly summarized.
4.    The section 6 is also mainly about the authors experiences instead of all findings. The ways of usage of stem cells for therapy should be clearly identified and described.
5.    Why the only bone marrow processing was identified in Fig.2? What about other stem cells, since it is a review?
6.    Section 7. Are results of clinical trials? If yes, the title could give this information. The statements „used a smaller amount of stem cells “ are not scientifically clear. The more detail description of clinical trials (cell amount, duration, ways of cell application and other) is required in order to compare the better and worse results.
7.    Section 8. The title - “….the best”. The section title does not correspond to its content. Mentioned only two cell types without answering to the title’s question.
8.    Section 9 is too short and not summarizing all results presented in this review. The statement that “cellular  therapy is a safe method for no-option CLTI patients...“ does not sound promising.
9.    Conclusions are also superficial.

Minor:
1.    Line 95 – too many “after”.
2.    Line 170 – “peripheral or umbilical cord blood...“

Comments on the Quality of English Language

The quality of English language is good.

Author Response

Thank you very much for taking the time to review this manuscript. Please find the detailed responses below with poitn-by-point responses, and the corresponding revisions with highlighted changes in the re-submitted files.

The manuscript needs some rearrangement and more clear and better description of findings, and their sumarry. Here are some remarks.

Major remarks: 

Comments 1: Fig. 1 would be more informative presenting known ways of CLTI treatment instead of mentioning possible cell types.

Response 1: We added the part with the source of stem cells to make the figure more informative. The process of stem cell therapy is described in Fig. 2.

Comments 2: The paragraphs 4 and 5 needs connection with further separation into subsections (4.1….; 4.2…) dedicated for each cell types used in the CLTI. The information also should be summarized in Table 1 with the citations identifying cell type, ways of application, final result, progress or other main findings important for the CLTI therapy. 

Response 2: We divided the section into 2 subsections - types of stem cells and stem cell subpopulations. The second one was divided into 4 other parts for better understanding. Table 1 was created as you suggested.

Comments 3: The section 5 is mainly only about the one cells type, ECFC. What about MAC cells? Only at the end some other cells were mentioned. The section title does not correspond to the content and is not clearly summarized. 

Response 3: As published in a review by Medina et al., MACs have only supportive or paracrine function in the mechanism of ACT. Later Chamber et al. described the possible negative effect of MACs in patients with diabetes, which we have discussed in the article. On the other hand, ECFCs are responsible for direct effect in ACT by vessel formation and it is the reason why I described this subpopulation more. Since the number of ECFCs in the collected material is low, in our opinion it is important to describe more possibilities for their enhancement. 

Comments 4: The section 6 is also mainly about the authors experiences instead of all findings. The ways of usage of stem cells for therapy should be clearly identified and described.

Response 4: We added the other section about stem cell therapy including the use of mesenchymal cells and described the process in accordance with other published studies.

Comments 5: Why the only bone marrow processing was identified in Fig.2? What about other stem cells, since it is a review? 

Response 5: We significantly changed the design of the figure, including also MSCs with different types of sources.

Comments 6: Section 7. Are results of clinical trials? If yes, the title could give this information. The statements „used a smaller amount of stem cells “ are not scientifically clear. The more detail description of clinical trials (cell amount, duration, ways of cell application and other) is required in order to compare the better and worse results. 

Response 6: We completely re-wrote Section 7 - it is now dedicated for “In-depth analysis of the most important clinical trials” and we critically assessed 5 important RCTs including Table 1 where we compared their characteristics.

Comments 7: Section 8. The title - “….the best”. The section title does not correspond to its content. Mentioned only two cell types without answering to the title’s question. 

Response 7: After a careful consideration we removed this section and the information was  divided into other sections.  

Comments 8: Section 9 is too short and not summarizing all results presented in this review. The statement that “cellular  therapy is a safe method for no-option CLTI patients...“ does not sound promising. 

Response 8: We added other studies which discuss the safety of ACT. In accordance with Global vascular guidelines from 2019 it is recommended therapy on the level 1B, so it is possible to consider this therapy safe. 

Comments 9: Conclusions are also superficial.

Response 9: For better understanding, we pointed out 3 main endpoints, which could have clinical impact for patients. 

Minor: 
Comments 1: Line 95 – too many “after”.
Response 1: Corrected.

Comments 2: Line 170 – “peripheral or umbilical cord blood...“
Response 2: Corrected.

Reviewer 2 Report

Comments and Suggestions for Authors

The manuscript provides a comprehensive review of stem cell therapies (ACT) for critical limb-threatening ischemia (CLTI). It outlines the types of stem cells used, including bone marrow-derived mononuclear cells (BM-MNCs), peripheral blood mononuclear cells (PB-MNCs), and mesenchymal stem cells (MSCs), and discusses their mechanisms of action and comparative efficacy. The review also touches on innovative and experimental approaches, such as genetic and nanoparticle-based therapies, and highlights various administration methods for stem cell treatments.

Specific comments on the manuscript:

1. In abstract, it is better to add which types of stem cells and techniques are compared in this review. This would provide a clearer understanding of the scope right from the abstract.

2. In line 44-49, could the manuscript explore in more detail why certain studies favor direct revascularization despite other research suggesting no significant difference? Are there any patient-specific factors or conditions that might tip the balance in favor of one approach over the other?

3. In line 66-70, the manuscript given that the evidence for prostanoids is described as insufficient, what are the reasons for their continued use? Is there ongoing research that could strengthen the case for or against prostanoids in treating CLTI?

4. In part 3 line 82, the section mentions innovative treatments at the genetic or nanoparticle level. Are these therapies at the clinical practice stage, or are they still primarily experimental? If they are not in clinical practice, why are they mentioned in this part?

5. In part 4, line 110, how do BM-MNCs and PB-MNCs compare in terms of efficacy and safety in ACT? Are there particular patient profiles where one type might be preferred over the other? Why they prefer to use these stem cells, is there any explanation for this?

6. In line 119, given that there are no studies on the use of neural stem cells (NSCs) in CLTI, what are the potential benefits and challenges of exploring this path? Could the successes seen in stroke patients provide a basis for similar studies in CLTI?

7. In line 155-159, how do the mechanisms of action of MACs and ECFCs differ, also what specific roles do each play in angiogenesis and tissue repair?

8. In line 159, what are the mechanisms by which MACs can become inflammatory in diabetic patients? How does this impact their overall effectiveness in ACT?

9. In line 188, can you provide more details on how pre-activation with vitamin D, integrin B1, Fucoidan, or ARA290 affects the functionality of ECFCs? What are the underlying biological processes that are enhanced by these treatments?

10. In line 197-200, is there any strategies are being explored to increase the number of ECFCs available for therapy, particularly in older patients or those with diabetes?

11. In line 246-250, how do intramuscular and intra-arterial injection methods compare in terms of patient outcomes? Are there specific conditions or patient factors that might make one method more preferable than the other?

12. In line 250, what are the challenges associated with systemic administration of stem cells, and how might these challenges be overcome?

13. In part 7 treatment results, it would be better to offer a more detailed comparison of results across studies, including sample sizes, methodologies, and specific outcomes.

14. In line 294-298, the section reports conflicting results in different studies, PROVASA and JUVENTAS. It would be helpful to explore potential reasons for these discrepancies, such as differences in patient demographics or treatment protocols.

15. In section 8 line310, it would be better to discuss the comparative effectiveness of these stem cells in more detail, like, how do BM-MNCs and PB-MNCs compare in terms of patient outcomes?

16. In part9, line 332, are there any rare or severe adverse effects that have been documented on long-term safety? Also, how are side effects managed in clinical practice? Are there any specific protocols or recommendations?

17. In conclusion, it is recommended to add more key insights from the review and their effects for clinical practice. Also, offer specific recommendations for future research, like areas that need further exploration or gaps that need to be talked.

Author Response

Thank you very much for taking the time to review this manuscript. Please find the detailed responses below with poitn-by-point responses, and the corresponding revisions with highlighted changes in the re-submitted files.

The manuscript provides a comprehensive review of stem cell therapies (ACT) for critical limb-threatening ischemia (CLTI). It outlines the types of stem cells used, including bone marrow-derived mononuclear cells (BM-MNCs), peripheral blood mononuclear cells (PB-MNCs), and mesenchymal stem cells (MSCs), and discusses their mechanisms of action and comparative efficacy. The review also touches on innovative and experimental approaches, such as genetic and nanoparticle-based therapies, and highlights various administration methods for stem cell treatments.

Specific comments on the manuscript:

Comments 1: In abstract, it is better to add which types of stem cells and techniques are compared in this review. This would provide a clearer understanding of the scope right from the abstract.

Response 1: We described types of the cells and also the methodology of ACT.

Comments 2: In line 44-49, could the manuscript explore in more detail why certain studies favor direct revascularization despite other research suggesting no significant difference? Are there any patient-specific factors or conditions that might tip the balance in favor of one approach over the other?  

Response 2: We added information about the choice between indirect (IR) and direct (DR) revascularization, that the presence of collaterals is important. Patients after IR with preserved collaterals have the same results in terms of wound healing than patients after DR.

Comments 3: In line 66-70, the manuscript given that the evidence for prostanoids is described as insufficient, what are the reasons for their continued use? Is there ongoing research that could strengthen the case for or against prostanoids in treating CLTI? 

Response 3: We agree that this part was very short, therefore we explained the decision for no indication of prostanoids for these patients until now. However, in accordance with recent study, the prostanoid treatment after PTA led to better effect on major amputation rate. This could be the future way for their use in CLTI.

Comments 4: In part 3 line 82, the section mentions innovative treatments at the genetic or nanoparticle level. Are these therapies at the clinical practice stage, or are they still primarily experimental? If they are not in clinical practice, why are they mentioned in this part? 

Response 4: We are aware of that the title of this section was’t correct, because we mentioned here the word “patients” and the methods are just experimental, but as we added to review -  it is the challenge for the future to modify it to clinical practice, and thus extend the possibilities for no-option patients.

Comments 5: In part 4, line 110, how do BM-MNCs and PB-MNCs compare in terms of efficacy and safety in ACT? Are there particular patient profiles where one type might be preferred over the other? Why they prefer to use these stem cells, is there any explanation for this? 

Response 5: The comparison between BM-MNCs and PB-MNCs is mentioned in the part about treatment results. We pointed out studies confirming or denying differences between these cell groups. 

Comments 6: In line 119, given that there are no studies on the use of neural stem cells (NSCs) in CLTI, what are the potential benefits and challenges of exploring this path? Could the successes seen in stroke patients provide a basis for similar studies in CLTI? 

Response 6: Neural stem cells are used in neurology in patients after stroke by regenerative nerve mechanisms. Most patients with CLTI also suffer from diabetic neuropathy. In this way, it is potentially possible to avoid the risk of losing a limb due to a defect condition or exacerbation of infection, which neuropathy also contributes to.

Comments 7: In line 155-159, how do the mechanisms of action of MACs and ECFCs differ, also what specific roles do each play in angiogenesis and tissue repair? 

Response 7: The mechanism and potential use of these subpopulations is described in the following subsections.

Comments 8: In line 159, what are the mechanisms by which MACs can become inflammatory in diabetic patients? How does this impact their overall effectiveness in ACT? 

Response 8: As published in a review by Medina, MACs have only supported or paracrine function in the mechanism of ACT. Later Chamber described the possible negative effect of MAC in patients with diabetes, which we have discussed in the article. MACs in diabetic patients also show higher inflammatory potential due to upregulation of Interleukin β (IL β).

Comments 9: In line 188, can you provide more details on how pre-activation with vitamin D, integrin B1, Fucoidan, or ARA290 affects the functionality of ECFCs? What are the underlying biological processes that are enhanced by these treatments? 

Response 9: We added more detailed information about the influence of above-mentioned molecules on ECFCs funcion.

Comments 10: In line 197-200, is there any strategies are being explored to increase the number of ECFCs available for therapy, particularly in older patients or those with diabetes? 

Response 10: As we mentioned in the article “Triggering factors that lead to ECFC dysfunction in patients with diabetes are hyperglycemia, oxidative stress and inflammation (Liu et al., 2024).” Since patients with CLTI are older and have a lot of comorbidities changing their oxidative stress is very difficult. However, inflammation for gangrene or chronic ulcer could be reduced by antibiotics and we should have tried for better glycemic control before stem cell treatment.

Comments 11: In line 246-250, how do intramuscular and intra-arterial injection methods compare in terms of patient outcomes? Are there specific conditions or patient factors that might make one method more preferable than the other? 

Response 11: Both intramuscular and intra-arterial administration have different advantages. As we added to the section, in general intramuscular injection is less invasive and harmful which is preferable in CLTI patients, on the other hand, the advantage of  intra-arterial method is to direct stem cells to peripheral areas with sufficient oxygen and nutrient supply to promote the activity of new cells (Yunir et al., 2021). Systemic administration of stem cells is less common for the difficulty of the administration and adverse events.

Comments 12: In line 250, what are the challenges associated with systemic administration of stem cells, and how might these challenges be overcome? [45]. 

Response 12: In our opinion, it is very questionable if the stem cells administered systemic route are able to transfer into the ischemic location with the significant effect and as we mentioned in the point 11, the administration is complicated with possible adverse events.

Comments 13: In part 7 treatment results, it would be better to offer a more detailed comparison of results across studies, including sample sizes, methodologies, and specific outcomes.

Response 13: We completely re-wrote Section 7 in accordance with your suggestions and also the comments of other reviewers. We chose 5 most important RCTs and critically assessed them including newly added Table 1 to compare them. We hope that this will improve our manuscript.

Comments 14: In line 294-298, the section reports conflicting results in different studies, PROVASA and JUVENTAS. It would be helpful to explore potential reasons for these discrepancies, such as differences in patient demographics or treatment protocols. 

Response 14: All these discrepancies are now explained in the new Section 7.

Comments 15: In section 8 line310, it would be better to discuss the comparative effectiveness of these stem cells in more detail, like, how do BM-MNCs and PB-MNCs compare in terms of patient outcomes? 

Response 15: After a careful consideration we removed this section and the information is divided into other sections.

Comments 16: In part9, line 332, are there any rare or severe adverse effects that have been documented on long-term safety? Also, how are side effects managed in clinical practice? Are there any specific protocols or recommendations? 

Response 16: We added other studies which discuss the safety of ACT. In accordance with Global vascular guidelines from 2019 it is recommended therapy on the level 1B, so it is possible to consider this therapy safe.

Comments 17: In conclusion, it is recommended to add more key insights from the review and their effects for clinical practice. Also, offer specific recommendations for future research, like areas that need further exploration or gaps that need to be talked. 

Response 17: For better understanding, we pointed out 3 main endpoints, which could have clinical impact for patients. 

Reviewer 3 Report

Comments and Suggestions for Authors

Author Response

Thank you very much for taking the time to review this manuscript. Please find the detailed responses below with poitn-by-point responses, and the corresponding revisions with highlighted changes in the re-submitted files.

Diabetic microangiopathy (for example foot ulcer) is a common complication of advanced diabetes mellitus affecting millions of people. The past few decades have seen significant advances in the management of these complications with improvement of quality of life. Stem cell therapy in particular local application of autologous hematopoietic stem cells represents a promising therapeutic choice for no-option patients. In this manuscript, the authors sought to review the advances in autologous stem cell therapy for diabetes associated peripheral ischemia. The manuscript starts with a brief review on alternative revascularization therapies for example extracorporeal shock-wave therapy (ESWT), transcatheter arterialization of deep veins or targeted therapy with small molecules. Subsequently, the authors reviewed the cellular sources that mediate therapeutic effect of autologous cell therapy (ACT) with a focus on ECFC (i.e. endothelial colony-forming cells) followed by a summary on treatment results of ACT. Overall, this manuscript lacks novel, systemic and in-depth discussion on critical aspects of the topic.

Comments 1: This reviewer does not see enough novelty in this manuscript as compared to recent review articles on this topic.

Response 1: In our opinion, this review is an interconnection between molecular and clinical aspects of autologous cell therapies. It brings some new insights about the question, e.g. section number 3. New promising options for revascularization. However, we tried to introduce information as clinicians with a close connection directly to patients. 

Comments 2: Professional language editing is encouraged to improve readability and correct the numerous grammatical errors.

Response 2: We tried to edit some mistakes and the manuscript was checked by a native English speaker.

Comments 3: This reviewer cannot see clear take-home message at the end of the manuscript or a brief and accurate summary for each section. For example, what’s the value of ACT for a given CLTI patient? In addition, part of the subtitles do not have interdependent connection with the contents and questions in these subtitles remain unanswered. For example, what is the best type of stem cells for ACT (section 8)? What cells mediate the therapeutic effect of ACT (section 5)?

Response 3: We changed the last paragraph of our manuscript to be briefer and crystal clear for the reader of IJMS. We focused on 3 main take-home messages for clinic and future research, i.e. the most advantageous type of cell for CLTI patients, the most preferred route of administration and the possibility for increasing the effect of ACT.

Comments 4: The ACT-associated contents (sections 4 – 9) should be grouped into one major section starting with an overview of the technology including (but not limited) the procedure (current section 6), mechanism of action, technical and practical considerations (for example cost and side effect), indications and contraindications. The following topics should also be included: 1) the status (advantages and disadvantages) of ACT compared to alternative revascularization methods, 2) the technological development of ACT with the improvement in clinical application, 3) the limitations of ACT and directions for future research?

Response 4: We reorganized the part about stem cell therapy and divided extra paragraphs of criteria, where we also mention criteria in other studies except for ours. Side effects and safety of ACT are described in the section “Is ACT a safe method?”, in which we have added new studies. Direction and main endpoints are mentioned in conclusion. We also added the study about the comparison between ACT and PTA in section 7. A new sentence about the price of cell therapy was added, but the price is very dependent on the chosen method. The price of Gelofusin method provided at our institute is aaproximately 1000 eures. 

Comments 5: There is lack of in-depth analysis for some critical large-scale clinical studies (milestones in the field). Instead, this manuscript describes a number of studies in parallel.

Response 5: We changed that and added an in-depth analysis for the most important studies in our opinion - JUVENTAS, PROVASA, RESTORE-CLI, PACE and study by Sharma et al. Furthermore, we added the table with an overview of these studies for better reading.

Comments 6: There are a number of inappropriate citations of references. For example, reference #46 is a meta-analysis but the citation gives an impression of an original article. Another example, reference #12 is a letter to the editor that questions the conclusion of the original research (described in lines 75 – 80). In this case, the original article should be cited. In addition, the narrative “we” in line 79 gives the impression that the study was conducted by the authors.

Response 6: We checked references, corrected the wrong ones and added new references in accordance with corrections from all reviewers, thank you for your suggestions.

Comments 7: Section 5 focuses on ECFC which is described in great details. Apparently contradictory, the authors provide evidence showing ECFC is irrelevant for ACT due to extremely low frequency.

Response 7: We re-wrote Section 5 in accordance with the comments of other reviewers. It is true that the frequency of ECFC is low, whereas other studies and reviews report that ECFCs have  potential vasoreparative potential in both PAD and retinopathy (O'Leary et al., 2019: doi: 10.1016/j.exer.2019.03.001) and this effect was described several times in the past. 

Comments 8: Please clarify the selection criteria in section 6 (lines 225- 234) if it’s institutional or widely accepted guidelines?

Response 8: These criteria are from our institution, we explained it in the text. However, we also added the paragraph about the criteria from other studies. 

Round 2

Reviewer 1 Report

Comments and Suggestions for Authors

The authors have addressed the comments and revised the article accordingly, resulting in significant improvements to the manuscript.

Author Response

Comments 1: The authors have addressed the comments and revised the article accordingly, resulting in significant improvements to the manuscript.

Response 1: Thank you very much for taking the time to review this manuscript. 

Reviewer 2 Report

Comments and Suggestions for Authors

The manuscript offers a strong review of ACT, particularly focusing on MSCs, but it would benefit from clearer structure and deeper exploration of recent advancements. The conclusions are well-founded, and with these revisions, the manuscript could significantly contribute to the field of regenerative medicine.

A more structured organization would improve readability, especially in sections like "Treatment Results," which occasionally lacks clear transitions between studies. Grouping the studies by cell types or routes of administration could make the narrative easier to follow.

Comments on the Quality of English Language

There are several grammatical issues that need to be addressed, and I recommend a thorough review by a native English speaker. Some of the key areas for improvement include:

  • The manuscript inconsistently shifts between past and present tense, disrupting the flow. Maintaining consistent verb tenses is crucial (e.g., "was" vs. "is" vs. "will be").
  • In lines 315-316, the sentence starts in the past tense ("was purified") but then switches to future tense ("will be seeded"). Since this is describing a completed procedure, it should use past tense consistently: "was seeded."
  • Redundant phrases should be eliminated, such as the repeated term “incubation incubation” on line 318.
  • Informal language, like “As I mentioned above” in line 305, should be replaced with objective phrasing, such as "As mentioned above," in keeping with the formal tone of scientific writing.
  • In line 293, "The second source of stem cells, which we used in our institute were PB-MNCs, but in definitely fewer patients," contains a grammatical error. Since "source" is singular, "were" should be changed to "was."
  • In line 298, the phrase "40-50 ml of the final product is administered intramuscularly" mixes present tense ("is administered") with surrounding past-tense descriptions. To maintain consistency, "is administered" should be changed to "was administered."

These revisions, along with a detailed grammatical review, would enhance the clarity and professionalism of the manuscript.

Author Response

Thank you very much for taking the time to review this manuscript. Please find the detailed

responses below and the corresponding revisions in the re-submitted files. 

Comments 0: The manuscript offers a strong review of ACT, particularly focusing on MSCs, but it would benefit from clearer structure and deeper exploration of recent advancements. The conclusions are well-founded, and with these revisions, the manuscript could significantly contribute to the field of regenerative medicine. 

Comments 1: A more structured organization would improve readability, especially in sections like "Treatment Results," which occasionally lacks clear transitions between studies. Grouping the studies by cell types or routes of administration could make the narrative easier to follow.

Response 1: For better clarity we created subsections and we wrote a brief summarization of the results of key studies.

Comments 2: Comments on the Quality of English Language

There are several grammatical issues that need to be addressed, and I recommend a thorough review by a native English speaker. Some of the key areas for improvement include:

  • The manuscript inconsistently shifts between past and present tense, disrupting the flow. Maintaining consistent verb tenses is crucial (e.g., "was" vs. "is" vs. "will be").
  • In lines 315-316, the sentence starts in the past tense ("was purified") but then switches to future tense ("will be seeded"). Since this is describing a completed procedure, it should use past tense consistently: "was seeded."
  • Redundant phrases should be eliminated, such as the repeated term “incubation incubation” on line 318.
  • Informal language, like “As I mentioned above” in line 305, should be replaced with objective phrasing, such as "As mentioned above," in keeping with the formal tone of scientific writing.
  • In line 293, "The second source of stem cells, which we used in our institute were PB-MNCs, but in definitely fewer patients," contains a grammatical error. Since "source" is singular, "were" should be changed to "was."
  • In line 298, the phrase "40-50 ml of the final product is administered intramuscularly" mixes present tense ("is administered") with surrounding past-tense descriptions. To maintain consistency, "is administered" should be changed to "was administered."

These revisions, along with a detailed grammatical review, would enhance the clarity and professionalism of the manuscript.

Response 2: We corrected grammatical issues mentioned in your comments. Moreover, the manuscript was checked by an English diabetologist who was also added as co-authors.

Reviewer 3 Report

Comments and Suggestions for Authors

The quality of the manuscript has been improved following rapid revision however some new issues arise and need to be addressed.

1) For in-depth review of critical studies, this reviewer feels it’s inappropriate to simply list literatures for the readers (page 10 - 11). There should be a more concise summary followed by discussion.

2) Part of the writing in section 5.3 (first paragraph) is more like presenting a protocol.

Comments on the Quality of English Language

There are still a number of grammatical errors. 

Author Response

Thank you very much for taking the time to review this manuscript. Please find the detailed

responses below and the corresponding revisions in the re-submitted files. 

Comments 0: The quality of the manuscript has been improved following rapid revision however some new issues arise and need to be addressed.

Comments 1: For in-depth review of critical studies, this reviewer feels it’s inappropriate to simply list literatures for the readers (page 10 - 11). There should be a more concise summary followed by discussion.

Response 1: For better clarity we created subsections and we wrote a brief summarization of the results of key studies.  

Comments 2: Part of the writing in section 5.3 (first paragraph) is more like presenting a protocol. 

There are still a number of grammatical errors. 

Response 2: We removed detailed information about the process of cultivation and shortened all sections to better fit to review.  

Comments 3: There are still a number of grammatical errors. 

Response 3: The manuscript was checked by an English diabetologist who was also added as co-authors.